# A New Model Algorithm for Estimating the Inhalation Exposure Resulting from the Spraying of (Semi)-Volatile Binary Liquid Mixtures (SprayEva)

**DOI:** 10.3390/ijerph192013182

**Published:** 2022-10-13

**Authors:** Martin Tischer, Jessica Meyer

**Affiliations:** BAuA: Federal Institute for Occupational Safety and Health, Unit Occupational Exposure, Friedrich-Henkel-Weg 1-25, 44149 Dortmund, Germany

**Keywords:** spraying, (semi)-volatile mixtures, evaporation models, mass balance, continuous application, extended Euler algorithm, backpressure, activity coefficients, occupational exposure, biocidal products

## Abstract

The spraying of liquid multicomponent mixtures is common in many professional and industrial settings. Typical examples are cleaning agents, additives, coatings, and biocidal products. In all of these examples, hazardous substances can be released in the form of aerosols or vapours. For occupational and consumer risk assessment in regulatory contexts, it is therefore important to know the exposure which results from the amount of chemicals in the surrounding air. In this research, a mechanistic mass balance model has been developed that covers the spraying of (semi)-volatile substances, taking into account combined exposure to spray mist, evaporation from droplets, and evaporation from surfaces as well as the nonideal behaviour of components in liquids and backpressure effects. For wall-spraying scenarios, an impaction module has been developed that quantifies the amount of overspray and the amount of material that lands on the wall. Mechanistically, the model is based on the assumption that continuous spraying can be approximated by a number of sequentially released spray pulses, each characterized by a certain droplet size, where the total aerosol exposure is obtained by summation over all release pulses. The corresponding system of differential equations is solved numerically using an extended Euler algorithm that is based on a discretisation of time and space. Since workers typically apply the product continuously, the treated area and the corresponding evaporating surface area grows over time. Time-dependent concentration gradients within the sprayed liquid films that may result from different volatilities of the components are therefore addressed by the proposed model. A worked example is presented to illustrate the calculated exposure for a scenario where aqueous solutions of H_2_O_2_ are sprayed onto surfaces as a biocidal product. The results reveal that exposure to H_2_O_2_ aerosol reaches relevant concentrations only during the spraying phase. Evaporation from sprayed surfaces takes place over much longer time periods, where backpressure effects caused by large emission sources can influence the shape of the concentration time curves significantly. The influence of the activity coefficients is not so pronounced. To test the plausibility of the developed model algorithm, a comparison of model estimates of SprayExpo, SprayEva, and ConsExpo with measured data is performed. Although the comparison is based on a limited number (N = 19) of measurement data, the results are nevertheless regarded as supportive and acceptable for the plausibility and predictive power of SprayEva.

## 1. Introduction

Room spraying as well as spraying onto rigid walls occurs in many industrial, professional, and consumer applications and can lead to high inhalation exposure to vapours and aerosols. At the same time, dermal exposure via the deposition of aerosol on the skin and via other pathways can also play a relevant role [1,2]. Hence, the components of spray solutions such as biocidal products, paints, cleaning agents, and air fresheners can be highly relevant for occupational as well as consumer risk assessment in regulatory contexts. In particular, the REACH Regulation [3], the Plant Protection Products Regulation (PPPR) [4], and the Biocidal Products Regulation (BPR) [5] should be mentioned here. Human exposure during spraying activities includes inhalation exposure to vapours as well as to low-volatility and (semi)-volatile substances released as aerosols. According to DIN EN 13936 [6], for substances with a vapor pressure at room temperature of less than 100 Pa and more than 0.001 Pa, sampling methods should be selected that record vapour and aerosol simultaneously.

Given the large number and variety of chemical substances present in spray products and the variability in use patterns, it is not possible to measure all potential exposures in different workplaces, but exposure modelling is imperative as well. This need has been met by the development of a variety of exposure models in research and regulatory communities [7]. Lists of models using different specific approaches to estimate exposure are also given in the BPR and REACH guidance [8,9] and by Hahn et al. [7]. These models are designed for different applications and different tiers of exposure assessment (i.e., screening to higher tiers), where a distinction is made between two sources of exposure: (1) primary exposure from the spray cloud and its overspray; (2) secondary exposure caused by the evaporation of substances from the treated surfaces.

Most of the currently available models focus on the assessment of primary exposure since the substance of interest in spray formulations is often nonvolatile (e.g., biocides), and therefore, (semi)-volatile components such as solvents are often not addressed further [10]. Available higher-tier models, e.g., the spray module of ConsExpo [11] and SprayExpo [1,12], do not account for secondary exposure due to the evaporation of hazardous substances from the spray cloud and sprayed surfaces. Nevertheless, there are a number of situations where secondary exposure must be considered. Typical examples include the area spraying of aqueous disinfectants and cleaning formulations containing (semi)-volatile components such as hydrogen peroxide, glutaraldehyde, and alcohols. Therefore, the extension of mechanistic model approaches is needed to cover the spraying of (semi)-volatile substances, i.e., combined exposure to spray mist, evaporation from droplets, and evaporation from surfaces.

Based on previous work [13], a new iterative model algorithm, SprayEva, is proposed that aims to include both room- and wall-spraying situations. The model is based on the assumption that continuous spraying can be approximated by a number of sequentially released spray pulses, each characterized by a certain droplet size, where the total aerosol exposure is obtained by summation over all release pulses [1]. After the release of the spray, the material is assumed to be dispersed immediately and homogeneously (completely mixed) through the room air. At the same time, aerosol droplets are assumed to be removed by ventilation and evaporation, leading to a release of vapour and the shrinking of the droplets. In addition, droplets are also removed from air as they drop to the floor due to gravitation. Evaporating material deposited on the floor then contributes to the overall vapour concentration as a secondary emission source. It has to be noted that according to Stokes’s law, the velocity with which droplets fall is determined by the size of the (shrinking) droplets (assuming spherical particles) and the droplet’s mass density. That is, due to evaporation shrinking, droplets consisting of volatile components tend to settle slower than nonvolatile droplets but also disappear faster by evaporation. This leads to higher vapour concentrations in the air.

In the case of wall spraying, only the overspray (the portion of the released spray that does not reach the wall but becomes airborne) acts as a primary emission source, while material that lands on the wall or on the floor evaporates there, acting as a secondary emission source. Since workers typically apply the product continuously the treated area and the evaporating surface area grows over time. Time-dependent concentration gradients within the sprayed liquid films that may result from different volatilities of the components are therefore addressed with the proposed model. Hence, the proposed modelling approach is quite flexible and reflects a wide variety of exposure situations and pattern related to the spraying of (semi)-volatile binary mixtures.

## 2. Methods

### 2.1. General Mass Balance Equations

In essence, the conceptual model for aerosol exposure described above can be expressed in mathematical terms by a set of mass balance differential equations. These mass balance equations are based on the assumption that continuous aerosol release can be subdivided into a number *N_l_* of pulses *l* released at discrete time steps and that overall exposure to aerosol droplets in a room with a volume *V_room_* results from summation over all release pulses with initial aerosol concentration *A_init,c_* and droplet size classes *c*, where each droplet size class is characterised by an initial droplet diameter *d_init,c_*. When the interval Δ*t_l_* between the pulses decreases, a quasi-continuous application can be achieved.

Taking into account the differential mass loss due to evaporation $\frac{dm_{ev}\left(t\right)}{dt}$, the mass loss due to settling $\frac{dm_{Fsed}\left(t\right)}{dt}$ and the mass loss due to ventilation $\frac{dm_{vent}\left(t\right)}{dt}\;$ as the relevant transport mechanisms, the differential change of the initial aerosol concentration $A_{l,c}\left(t_{l}\right)$ in the air of a single release pulse *l* can be expressed as:(1)$$\frac{dA_{room,l,c}\left(t\right)}{dt}=\frac{1}{V_{room}}\left(-\frac{dm_{vent,l,c}\left(t\right)}{dt}-\frac{dm_{Fsed,l,c}\left(t\right)}{dt}-\frac{dm_{ev,l,c}\left(t\right)}{dt}\right)$$

Given that (numerical) solutions to Equation (1) are available (which will be demonstrated in Section 2.2 and Section 2.3), the total aerosol exposure at the time *t* is then obtained by summation over all release pulses *N_l_* and droplet size classes *N_c_*:(2)$$A_{room}\left(t\right)=\sum_{c=1}^{N_{c}}\sum_{l=1}^{N_{l}}A_{room,l,c}\left(t\right)$$

Analogously, the differential change $\frac{dC_{room,i}}{dt}$ in the vapour concentration of a component *i* can be expressed taking into account the vapour mass loss rate of component *i* due to ventilation $Q_{vent}\cdot C_{room,i}\left(t\right)$ and the emission rates resulting from aerosol evaporation $\frac{dm_{Aev,i}\left(t\right)}{dt}$ and eventually from evaporation from the floor $\frac{dm_{Fev,i}\left(t\right)}{dt}$ and from the wall $\frac{dm_{Wev,i}\left(t\right)}{dt}$. The term *C_vent,i_* describes the vapour concentration of component *i* in the supply air and $Q_{vent}$ the ventilation rate.
(3)$$\frac{dC_{room,i}\left(t\right)}{dt}=\frac{1}{V_{room}}\cdot \left(\frac{dm_{Aev,i}\left(t\right)}{dt}+\frac{dm_{Wev,i}\left(t\right)}{dt}+\frac{dm_{Fev,i}\left(t\right)}{dt}\right)-\frac{Q_{vent}}{V_{room}}\cdot C_{room,i}\left(t\right)+\frac{Q_{vent}}{V_{room}}\cdot C_{vent,i}$$

From these mass balance considerations, it is clear that the physical processes behind the emission and mass loss rates are key to understanding the time dependency of aerosol and vapour exposure. Therefore, the corresponding processes will be described in more detail in Section 2.4, Section 2.5, Section 2.6, Section 2.7, Section 2.8 and Section 2.9.

### 2.2. Discretisation Approach

As the mass balance for a single release pulse can be described by a set of only two time-varying differential equations (Equations (1) and (3)) for each of the involved compounds, solving this system of ordinary differential equations (ODEs) analytically seems to be feasible at first sight. On the other hand, the calculation of the total exposure requires summation over all release pulses, which prevents analytical solutions to this problem. Hence, in order to obtain approximate time-dependent solutions to Equations (1) and (3), an extended Euler method was used that was developed in previous work [13].

In the case of wall spraying, a moving spray cone is assumed that sweeps a certain surface area *W* during the application. Hence, the treated surface area and the corresponding emission rate increase over time. As the sprayed substances may differ in their volatility, spatial concentration gradients within the sprayed liquid film on the surface can occur. In order to address this problem, we break down a certain area *W* into a sequence of *N_l_* small area elements ΔW that are numbered consecutively using index *l*. That is, the size of $\Delta W=W/N_{l}$ determines the spatial resolution of the proposed approach. We assume that fractions of the product are applied instantaneously, spray pulse by spray pulse, where the applied amount reflects the work rate of the worker. However, in principle, this approach can also take arbitrary values for the applied amount, or even zero to reflect periods when the worker stops the application (breaks, postapplication phase).

The next step in the discretisation procedure is to define the temporal resolution that is needed to adequately describe the time-dependent aerosol and vapour concentrations resulting from each release pulse. This can be accomplished by dividing the overall exposure time *t_expo_* into *N*_t_ time steps so that Δ*t = t_expo_/N_t_*. In brief, the time Δ*t_l_* between two release pulses *l* can be the same as Δ*t*. However, for reducing computational time, it may also be useful and justifiable to reduce the number of release pulses *N_l_* by allowing Δ*t_l_* to adopt integer multiples *IM* of Δ*t*, that is Δ*t_l_ =* Δ*t·IM* and *N*_t_
*=*
*N*_l_·*IM*. For clarification, in the results chapter, the value of *IM* will be stated explicitly for the example calculations.

In Figure 1, the discretisation procedure is exemplified for aerosol concentration *A* assuming *IM = 1*. The discretisation of the related variables of interest (droplet diameter, mole fractions, vapour concentration) and their substance-specific time-dependent derivates is not shown but can be expressed analogously. For each time step *k* and each release pulse *l*, the aerosol concentration at the beginning of time step *k* is represented by $A_{l,c}\left(t_{k}\right)$. For instance, $A_{2,1}\left(t_{3}\right)$ means the aerosol concentration of release pulse 2 in droplet size class 1 at the beginning of time step 3. The diagonal elements (that is *k = l*) of the triangular matrix in Figure 1 represent initial aerosol concentrations resulting from release pulse *l* at the time *t_k=l_ =* Δ*t (k* − 1*).* The temporal iteration process starts with time step *k =* 1 and proceeds successively for each release pulse *l* to the maximum number of time steps *N**_t_*.

That is, each release pulse *l* of droplet size class *c* is assumed to instantaneously apply an initial amount to the room volume at time $t_{k=l}$ leading to an initial aerosol concentration $A_{l,c}\left(t_{k=l}\right)=\frac{1}{V}\cdot \frac{dm_{l,c}\left(t_{k=l}\right)}{dt}\cdot \Delta t_{l}$. Correspondingly, the initial release rate of the spray is then $\frac{dm_{l,c}\left(t_{k=l}\right)}{dt}=\frac{A_{l,c}\left(t_{k=l}\right)}{\Delta t_{l}}\;V$. For wall spraying, the term $\frac{dm_{l,c}\left(t_{k=l}\right)}{dt}$ has to be replaced by the release rate of the overspray $\frac{dm_{os,l,c}\left(t_{k=l}\right)}{dt}$ (see Section 2.8).

Since all equations specified above refer to a droplet size class *c*, we now need to define the overall distribution that includes all the droplet size classes. Although there is no fundamental theoretical reason why droplet size data should approximate the lognormal distribution, it has been found to apply to most single-source aerosols [14]. Assuming a lognormal droplet size distribution with median mass diameter *d_m_* and geometric standard deviation *GSD*, a number of *N_c_* droplet size classes are defined, each in turn specified by a median mass diameter *d_m,c_*. The borders of each droplet size class *c* with median mass diameter *d_m,c_* are chosen in a way that it includes an equal proportion of $1/N_{c}$ of the total mass release rate $rr=\frac{dm_{l}\left(t_{k=l}\right)}{dt}$ of the spray. Then, the release rate for each droplet size class is $\frac{dm_{l,c}\left(t_{k=l}\right)}{dt}=\frac{1}{N_{c}}\cdot \frac{dm_{l}\left(t_{k=l}\right)}{dt}$. In turn, $A_{l,c}\left(t_{k=l}\right)=\frac{1}{V}\cdot \frac{1}{N_{c}}\cdot \frac{dm_{l}\left(t_{k=l}\right)}{dt}\cdot \Delta t_{l}$, describing the initial aerosol concentration of a spray pulse. An example of a cumulative mass distribution assuming lognormality is shown in Figure 2 with *d_m_* = 100 µm, *GSD* = 2 and *N_c_* = 7. The mass mediansc *d_m,c_* and boundaries of each droplet size class are depicted as green and black dots, respectively, each representing 1/*N_c_* of the total released mass. The symbol CDF denotes the cumulative mass distribution function.

### 2.3. Numerical Approximation Using Euler’s Method

To approximate the time-dependent solutions of the system of ordinary differential equations (ODE) described above and more specified in Section 2.4, Section 2.5, Section 2.6, Section 2.7, Section 2.8 and Section 2.9, the explicit Euler method is used. Although the Euler method is straightforward and hence not very precise [15], it is regarded sufficient as in occupational exposure modelling, high numerical precision is not required due to other sources of uncertainty in the exposure assessment process. Starting with the initial value *y(t*_0_*) = y*_0_, the Euler method iterates approximate solutions using the equation y(tk+1)=y(tk)−Δt·dy(tk)dt where tk=(k−1)·Δt with k∈ℕ, k=1,…Nt. The formulas of Euler’s algorithm for a single ODE do not change when the method is used to solve a system of ODEs. Hence, the iteration formula applied to Equation (1) for a release pulse *l* and droplet size class *c* for instance gives
(4)Aroom,l,c(tk+1)=Aroom,l,c(tk)−Δt dAroom,l,c(tk)dt

The iteration process over time for each release pulse *l* of droplet size class *c* at the time *t_k=l_* starts with the initial aerosol concentration *A_l,c_*(*t_k=l_*). It should be noted that the initial value can change over time, so that the modelling of intermittent application patterns is possible. The total aerosol concentration is obtained by summation over all release pulses and droplet size classes:(5)$$A_{room}\left(t_{k}\right)=\sum_{c=1}^{N_{c}}\sum_{l=1}^{N_{l}}A_{room,l,c}\left(t_{k}\right)$$

The substance-specific aerosol concentration can be calculated by multiplying the right side of Equation (5) with the mass fraction $w_{i,l,c}\left(t_{k}\right)$ of the respective component *i*:(6)$$A_{room,i}\left(t_{k}\right)=\sum_{c=1}^{N_{c}}\sum_{l=1}^{N_{l}}A_{room,l,c}\left(t_{k}\right)\cdot w_{i,l,c}$$
where
(7)$$w_{i,l,c}\left(t_{k}\right)=\frac{x_{i,l,c}\left(t_{k}\right)\cdot M_{i}}{\sum_{j=1}^{N_{j}}(x_{i,l,c}\left(t_{k}\right)\cdot M_{j})}$$
and
(8)$$x_{i,l,c}\left(t_{k}\right)=\frac{\frac{m_{i,l,c}\left(t_{k}\right)}{M_{i}}}{\sum_{j=1}^{N_{j}}\frac{m_{j,l,c}\left(t_{k}\right)}{M_{j}}}$$

$M_{i}$ denotes the molecular weight of component *i* and $x_{i,l,c}$ the mole fraction of component *i* in the droplet size class *c* of the aerosol. In terms of exposure, it is important to note that not all sizes of droplets suspended in the air can be breathed in with 100% efficiency. Thus, any exposure assessment should focus on the particles that can actually enter the respiratory tract. The International Standards Organisation (ISO) [16] has agreed upon a so-called inhalable fraction. For wind velocities less than 4 m/s, the inhalable fraction is given by
(9)$$I\left(d_{m,c}\right)=0.5\cdot \left(1+e^{-0.06\cdot \;d_{m,c}\cdot 10^{6}}\right)$$

The inhalable fraction of the total aerosol concentration is then obtained by multiplying the right sides of Equations (5) and (6) with the right side of Equation (9).
(10)$$A_{inh}\left(t_{k}\right)=\sum_{c=1}^{N_{c}}\sum_{l=1}^{N_{l}}A_{room,l,c}\left(t_{k}\right)\cdot I\left(d_{m,c}\right)$$

The other variables of interest (*x_i_*, *d*, and *C_i_*) can be obtained analogously by taking into account the mass losses of each of the three relevant mass transport processes (ventilation, evaporation, sedimentation). The corresponding mass balance equations needed for the iteration procedure will be described in the following.

It should be noted that the programming of the extended Euler algorithm of SprayEva is straightforward, essentially using three Do loops nested over *k*, *l*, *c*. In the Appendix A, a piece of pseudocode based on Mathematica^®^ is given that is intended to illustrate how the iteration algorithm works for the room spraying of a binary mixture.

### 2.4. Mass Loss Due to Ventilation

In the well-mixed room (one-box) model, the sprayed material is assumed to be dispersed immediately and homogeneously through the room air [17,18,19]. At first sight, this assumption seems to be too simplistic given the fact that during spraying, the concentration of the sprayed material is highest near the source and forms a concentration gradient throughout the room. On the other hand, air turbulence is generated by the momentum flux of the sprayed aerosol and ventilation air as well as thermal convection. Thus, homogenisation inside a room can be achieved in minutes [20] as long as the room volumes are not too large and the room is well mixed due to either natural or technical ventilation. Koch et al. [12] concluded from a number of model calculations and comparison with measured data that the average aerosol concentration in smaller rooms (<150 m³), can be described sufficiently well.

From a mathematical point of view the mass loss rate due to ventilation $\frac{dm_{vent,l,c}\left(t_{k}\right)}{dt}$ for an aerosol release pulse *l* of droplet size class *c* at time $t_{k}$ can be expressed in quite simple terms [18]:(11)$$\frac{dm_{vent,l,c}\left(t_{k}\right)}{dt}=Q_{vent}\cdot A_{l,c}\left(t_{k}\right)$$

Analogously, the differential change in the overall vapour concentration $C_{room,i}\left(t_{k}\right)$ of component *i* at time $t_{k}$ due to mass loss by ventilation is determined by the term $Q_{vent}\cdot C_{room,i}\left(t_{k}\right)$. This simplification is justifiable when room volumes are not too large and when the room is well mixed due to either natural or technical ventilation [19,21].

### 2.5. Mass Loss Due to Droplet Settling

The vertical motion of the droplets and their deposition onto horizontal surfaces is determined by settling in the earth’s gravity field. The velocity *v_sed_* at which droplets fall to the floor is given by Stokes’s settling velocity:(12)$$v_{sed}=\frac{g\cdot \rho_{dr}\left(t\right)\cdot d{\left(t\right)}^{2}}{18\cdot \eta }$$

This formula applies to droplets with diameters up to 50 µm. For larger droplets, a correction is necessary [14]. The changing droplet diameter d(t) and density $\rho_{dr}\left(t\right)$ of the droplet due to evaporation make the settling velocity a time-dependent quantity decreasing with increasing residence time, where g denotes the gravitational constant and η the air viscosity. The mass density of the droplet $\rho_{dr,l,c}\left(t\right)$ can be calculated using the liquid molar volumes $V_{i}$ of the pure components and their mole fractions *x_i_*, where $N_{i}$ denotes the total number of components:(13)$$\rho_{dr,l,c}\left(t\right)=\frac{\sum_{i=1}^{N_{i}}(M_{i}\cdot x_{i,l,c}\left(t\right))}{\sum_{i=1}^{N_{i}}(V_{i}\cdot x_{i,l,c}\left(t\right))}$$

Taking into account the surface area of the floor, *F*, the mass loss rate $\frac{dm_{Fsed,l,c}\left(t_{k}\right)}{dt}$ due to sedimentation for an aerosol release pulse *l* of droplet size class *c* at time $t_{k}$ can then be expressed as:(14)$$\frac{dm_{Fsed,l,c}\left(t_{k}\right)}{dt}=\frac{F\cdot g\cdot \rho_{dr,l,c}\left(t_{k}\right)\cdot A_{l,c}\left(t_{k}\right)\cdot d_{l,c}{\left(t_{k}\right)}^{2}}{18\cdot \eta }$$

The substance-specific settling rates $\frac{dm_{Fsed,i,l,c}\left(t\right)}{dt}$ can be obtained by multiplying the right side of Equation (14) with the mass fraction $w_{i}\left(t\right)$ (see Equation (7)) of the respective component *i*:(15)$$\frac{dm_{Fsed,i,l,c}\left(t_{k}\right)}{dt}=\frac{F\cdot g\cdot \rho_{dr,l,c}\left(t_{k}\right)\cdot A_{l,c}\left(t_{k}\right)\cdot d_{l,c}{\left(t_{k}\right)}^{2}}{18\cdot \eta }\cdot w_{i,l,c}\left(t_{k}\right)$$

The summation of the sedimentation flow of component *i* over all release pulses *l* and droplet size classes *c* at the time $t_{k}$ then results in
(16)$$\frac{dm_{Fsed,i}\left(t_{k}\right)}{dt}=\sum_{c=1}^{N_{c}}\sum_{l=1}^{N_{l}}\frac{dm_{Fsed,i,l,c}\left(t_{k}\right)}{dt}$$

The left side of Equation (16) denotes the total mass flow of component *i*, depositing on and potentially evaporating from the floor (see Section 2.7).

### 2.6. Evaporation of Two-Component Droplets

The evaporation of the (semi)-volatiles from the liquid phase, i.e., the vapour-droplet partitioning, depends on their effective equilibrium vapour pressures, which are determined by the chemical compositions and thus the concentrations and activity coefficients of the constituents in the spray formulation. Commercial preparations are often complex mixtures of many substances with different physical properties. In addition, a wide variety of use scenarios is possible, making the evaporation of multicomponent droplets a complex process. Hence, for a basic understanding of multicomponent droplet evaporation, it is necessary to investigate droplet evaporation using simplified models. A number of models have been proposed for the evaporation of substances from aerosol droplets, each of which makes different assumptions and simplifications [22]. This study is based on the simplified rapid-mixing model (SRMM) for two-component droplet evaporation that was suggested by Wilms [22]. It is called a rapid-mixing model because it assumes a rapid mixing of the different liquids so that there is no concentration gradient within the droplet. The following differential equations are given for the evaporation rate, the size of a droplet, and its composition for binary mixtures. By extension to Wilms, vapour concentrations *C_room_* far away from the droplet are not assumed to be zero but can take finite values. This is regarded as an important modification as backpressure effects may influence the evaporation significantly. The evaporation mass flux of a single droplet then results for a component *i* in
(17)$$\frac{dms_{ev,i}\left(t\right)}{dt}=2\cdot \pi \cdot d\left(t\right)\cdot D_{i}\cdot \left(C_{room,i}\left(t\right)-\frac{M_{i}\cdot p_{i}^{*}\cdot x_{i}\left(t\right)\cdot \gamma_{i}\left(x_{i}\left(t\right)\right)}{R\cdot T}\right)$$
where *d* is the droplet diameter, $D_{i}$ the effective binary diffusion coefficient of the component *i* in the air, *M_i_* the molecular weight, $p_{i}^{*}$ the saturation vapour pressure, *x_i_* the molar fraction, and $\gamma_{i}$ the activity coefficient of component *i*. The activity coefficient $\gamma_{i}$, which depends on the molar fraction of component *i*, was introduced in Equation (17) to account for nonideal mixtures. Due to the consumed heat of evaporation, the effective temperature of the droplets is lower than the air temperature. Therefore, a reference temperature *T_r_* was used in this work (for Equation (17)) approximated using Hubbard’s 1/3 rule $T_{r}=T_{dr}+1/3\cdot \left(T_{room}-T_{dr}\right)$ [22,23], where $T_{dr}$ is the temperature at the droplet surface, estimated according to Hinds [14], who proposed an empirical equation for water droplets, and $T_{room}$ the room temperature.

Equation (17) only describes the mass flow of a single droplet and needs to be extended for the calculation of a spray pulse probably consisting of thousands of droplets. For this purpose, it is assumed that the droplets are spherical and noninteracting and that the number of droplets *N_dr_* in the room volume is related to the aerosol concentration *A(t)* and the density of the droplet $\rho_{dr}$ via the following equation:(18)$$N_{dr}=\frac{6\cdot A\left(t\right)\cdot V_{room}}{\pi \cdot \rho_{dr}\cdot d{\left(t\right)}^{3}}$$

Assuming a two-component droplet (*N_i_* = 2) and combining Equations (13), (17), and (18), the following expression can be derived for the vapour emission of a release pulse *l* of component 1 at the time $t_{k}$:(19)$$\frac{dm_{ev,1,l,c}\left(t_{k}\right)}{dt}=\frac{12\cdot D_{1}\cdot V_{room}\cdot A_{l,c}\left[t_{k}\right]\cdot \left(C_{room,1}\left(t_{k}\right)-\frac{M_{1}\cdot p_{1}^{*}\cdot \gamma_{1}\left(x_{1,l,c}\left(t_{k}\right)\right)\cdot x_{1,l,c}\left(t_{k}\right)}{R\cdot T}\right)}{d_{l,c}{\left(t_{k}\right)}^{2}\cdot \rho_{dr,l,c}\left(t_{k}\right)}$$

The emission rate of the second component can then be determined by rearranging Equation (19) using the equation $x_{1}+x_{2}=1$.
(20)$$\frac{dm_{ev,2,l,c}\left(t_{k}\right)}{dt}=\frac{12\cdot D_{2}\cdot V_{room}\cdot A_{l,c}\left[t_{k}\right]\cdot \left(C_{room,2}\left(t_{k}\right)-\frac{M_{2}\cdot p_{2}^{*}\cdot \gamma_{2}\left(1-x_{1,l,c}\left(t_{k}\right)\right)\cdot (1-x_{1,l,c}\left(t_{k}\right))}{R\cdot T}\right)}{d_{l,c}{\left(t_{k}\right)}^{2}\cdot \rho_{dr,l,c}\left(t_{k}\right)}$$

The summation of the evaporation flow of component *i* over all release pulses *l* and droplet size classes *c* at the time $t_{k}$ results in
(21)$$\frac{dm_{ev,i}\left(t_{k}\right)}{dt}=\sum_{c=1}^{N_{c}}\sum_{l=1}^{N_{l}}\frac{dm_{ev,i,l,c}\left(t_{k}\right)}{dt}$$

Since Equations (19) and (20) include $d_{l,c}\left(t\right)$ and $x_{1,l,c}\left(t\right)$ as unknown functions, the set of equations needs to be extended with corresponding differential equations for $\frac{dd_{l,c}\left(t\right)}{dt}$ and $\frac{dx_{1,l,c}\left(t\right)}{dt}$. Following Wilms [22] (chapter 3.6) and using the simplified rapid-mixing model (SRMM) for two-component droplets, the following equations are obtained, where $V_{i}$ denotes the molar volume of component *i* and $\phi_{i}=D_{i}/\left(R\cdot T\right)$:(22)$$\begin{array}{c} \frac{dd_{l,c}\left(t_{k}\right)}{dt}=\frac{4}{d_{l,c}\left(t_{k}\right)}\cdot (C_{room,1}\left(t_{k}\right)/M_{1}\cdot R\cdot T\cdot V_{1}\cdot \phi_{1}+C_{room,2}\left(t_{k}\right)/M_{2}\cdot R\cdot T\cdot V_{2}\cdot \phi_{2}-p_{1}^{*}\cdot V_{1}\cdot \\ \gamma_{1}\left(x_{1,l,c}\left(t_{k}\right)\right)\cdot \phi_{1}\cdot x_{1,l,c}\left(t_{k}\right)-p_{2}^{*}\cdot V_{2}\cdot \gamma_{2}\left(1-x_{1,l,c}\left(t_{k}\right)\right)\cdot \phi_{2}\cdot \left(1-x_{1,l,c}\left(t_{k}\right)\right))\; \end{array}$$
(23)$$\begin{array}{c} \frac{dx_{1,l,c}\left(t_{k}\right)}{dt}=\frac{12}{d_{l,c}{\left(t_{k}\right)}^{2}}\cdot \left(-V_{2}+\left(V_{2}-V_{1}\right)\cdot x_{1,l,c}\left(t_{k}\right)\right)\cdot (C_{room,1}\left(t_{k}\right)/M_{1}\cdot \phi_{1}\cdot R\cdot T\cdot \left(x_{1,l,c}\left(t_{k}\right)-1\right)\\ +(C_{room,2}\left(t_{k}\right)/M_{2}\cdot \phi_{2}\cdot R\cdot T-\left(\phi_{1}\cdot p_{1}^{*}\cdot \gamma_{1}(x_{1,l,c}\left(t_{k}\right))-\phi_{2}\cdot p_{2}^{*}\cdot \gamma_{2}(1-x_{1,l,c}\left(t_{k}\right))\right)\\ \cdot \left(x_{1,l,c}\left(t_{k}\right)-1\right))\cdot x_{1,l,c}\left(t_{k}\right)) \end{array}$$

Given that (numerical) solutions to Equation (23) are available, the mole fraction of the second component can be determined via the equation $x_{2}=1-x_{1}$.

### 2.7. Evaporation from the Floor

The vertical motion of droplets in the earth’s gravity field leads to a mass flow of (semi)-volatile components potentially depositing on the floor as a liquid film. At the same time, evaporation from the liquid film occurs leading to a mass loss of the film by vapour emission. Hence, both processes have to be taken into account for deriving the differential mass change $\frac{dm_{F,i}\left(t_{k}\right)}{dt}$ of a component *i* in the liquid film on the floor at the time $t_{k}$:(24)$$\frac{dm_{F,i}\left(t_{k}\right)}{dt}=\frac{dm_{Fsed,i}\left(t_{k}\right)}{dt}-\frac{dm_{Fev,i}\left(t_{k}\right)}{dt}$$

While the sedimentation flow $\frac{dm_{Fsed,i}\left(t\right)}{dt}$ is governed by Equation (16), the evaporation rate $\frac{dm_{Fev,i}\left(t\right)}{dt}$ of a component *i* released at time $t_{k}$ from the liquid film can be predicted using the model suggested by Gmehling and Weidlich [24,25]:(25)$$\frac{dm_{Fev,i}\left(t_{k}\right)}{dt}=F\cdot \beta_{i}\cdot \left(\frac{M_{i}\cdot p_{i}^{*}\cdot x_{F,i}\left(t_{k}\right)\cdot \gamma_{i}\left(x_{F,i}\left(t_{k}\right)\right)}{R\cdot T}-C_{room,i}\left(t_{k}\right)\right)$$

In this model, it is assumed that evaporation is driven by the difference between the partial vapour pressure of an individual component *i* and its vapour pressure (backpressure) in the room air $p_{room,i}=C_{room,i}\cdot R\cdot T/M_{i}$, which is reflected in Equation (25) by the term $C_{room,i}\left(t_{k}\right)$. The evaporation rate of substance *i*, $\frac{dm_{Fev,i}\left(t\right)}{dt}$, is proportional to this pressure difference and depends on the surface area *F* of the mixture and the mass transfer coefficient *ß_i_*. The activity coefficient *γ_i_* as a function of $x_{F,i}\left(t_{k}\right)$ is used to take into account nonideal behaviour of the components in the liquid as defined by Raoult and Henry. The mass transfer coefficient *ß_i_* is a function of the diffusivity $D_{i}$ of the substance in the air and of the air flow *v*_air_ over the product surface [24]:(26)$$\beta_{i}=0.011\cdot \frac{v_{\mathrm{air}}^{0.96}\cdot \;D_{i}^{0.19}{}^{\;}}{\nu^{0.14}\cdot X^{0.04}}$$

Parameter notation and corresponding symbols and units are listed in Table 1. Previous work has described this modelling approach in more detail [13].

Combining Equations (24) and (25) then gives the differential mass change $\frac{dm_{F,i}\left(t_{k}\right)}{dt}$ of component *i* in the liquid film on the floor at the time $t_{k}$:(27)$$\frac{dm_{F,i}\left(t_{k}\right)}{dt}=\frac{dm_{Fsed,i}\left(t_{k}\right)}{dt}-F\cdot \beta_{i}\cdot \left(\frac{M_{i}\cdot p_{i}^{*}\cdot x_{F,i}\left(t_{k}\right)\cdot \gamma_{i}\left(x_{F,i}\left(t_{k}\right)\right)}{R\cdot T}-C_{room,i}\left(t_{k}\right)\right)$$
where the mole fraction of component *i* is defined as
(28)$$x_{F,i}\left(t_{k}\right)=\frac{\frac{m_{F,i}\left(t_{k}\right)}{M_{i}}}{\sum_{j=1}^{N_{j}}\frac{m_{F,j}\left(t_{k}\right)}{M_{j}}}$$

Since Equations (25) and (27) are coupled by Equation (28), the evaporation rate $\frac{dm_{Fev,i}\left(t_{k}\right)}{dt}$ and the differential mass change $\frac{dm_{F,i}\left(t_{k}\right)}{dt}$ of component *i* can be calculated simultaneously. In the case that evaporation is faster than sedimentation, potentially leading to numerical results where $m_{F,i}\left(t_{k}\right)\leq 0$, the evaporation flow is assumed to be determined by the sedimentation flow, that is, $\frac{dm_{Fev,i}\left(t_{k}\right)}{dt}=\frac{dm_{Fsed,i}\left(t_{k}\right)}{dt}$. In other words, the droplets are assumed to evaporate instantaneously when reaching the floor.

### 2.8. Impaction Module

For spraying on surfaces, only the nonimpacting fraction of the droplet spectrum (overspray) is relevant for aerosol exposure. At the same time, the fraction of the spray deposited on surfaces can evaporate and can contribute significantly to the overall vapour exposure. The importance of spray penetration and impaction is well recognized and has been extensively studied experimentally and theoretically [26]. A rigorous theory of spray dynamics would be very complex, making a self-consistent modelling of these processes a major challenge. However, an in-depth understanding of all involved processes is not always essential as long as modelling arrives at results within the required accuracy boundaries. On many occasions, it is therefore justifiable to develop simplified models suitable for practical applications. Based on the work of Sazhin et al. [27], Medrano et al. [28], and Su and Yao [29], the following simplifying assumptions are made in this study for estimating the proportion of the overspray *PO* and the (complementary) proportion (1 − *PO*) of the spray deposited on surfaces.

-The initial droplet velocity *v_*0*_* is equal to the velocity of the liquid at the injector tip.-A full-cone turbulent round jet ensuing from a circular nozzle is assumed.-Air entrainment beyond the vicinity of the nozzle is considered.-The droplet size distribution is constant during the transport process.-The approach is restricted to medium volatile mixtures that should not exceed the vapour pressure of water.

According to Su and Yao [29], an inertial impaction parameter *K* for impacting sprays is defined in terms of the momentum conservation to reveal the relationship between spray transportation and aerodynamic conditions. The impaction parameter is defined in Equation (29) as the quotient of the stopping distance λ(x) based on the flow mean velocity $\bar{v}\left(x\right)$ to the spray diameter D(x) at the cross section of the target plate. The symbol $\rho_{dr}$ denotes the droplet density, η the dynamic viscosity of the air, and $\bar{v}\left(x\right)$, the flow mean velocity of the spray as a function of the impacting distance *s*. The droplet diameter *d* is assumed to be constant within the flying time and thus can be set equal to the initial droplet diameter *d_init,c_*, belonging to droplet size class *c*. These are simplifying assumptions, but given that the flying time of larger droplets that may reach the surface is much shorter than their evaporation time, they are probably reasonable [1]. In addition, smaller droplets predominantly belong to the overspray, where they evaporate, contributing not to the impacted mass but to vapour exposure.
(29)$$K=\frac{\lambda \left(s\right)}{D\left(s\right)}=\frac{1}{18}\cdot \frac{\bar{v}\left(s\right)\cdot \rho_{dr}\cdot d^{2}}{D\left(s\right)\cdot \eta }$$

In order to obtain the flow mean velocity $\bar{v}\left(s\right)$ needed in Equation (29) Sazhin et al. [27] and Medrano et al. [28] propose a two-phase-fluid model for a full-cone turbulent round jet with only the apex angle θ of the cone being a disposable parameter (see Figure 3).

As a result, the dynamics of both droplets and entrained air can be described in terms of a two-phase flow with a zero relative velocity $\bar{v}\left(s\right)$ between air and droplets depending on the distance from the nozzle, *s*. Here, $v_{0}$ denotes the initial droplet velocity at the nozzle.
(30)$$\bar{v}\left(s\right)=\frac{2\cdot v_{0}}{1+\sqrt{1+\frac{4\cdot \rho_{air}}{\rho_{dr}}+\frac{16\cdot s\cdot \rho_{air}\cdot Tan\left(\theta \right)}{d\cdot \rho_{dr}}+\frac{16\cdot s^{2\cdot }\rho_{air}\cdot Tan{\left(\theta \right)}^{2}}{d^{2}\cdot \rho_{dr}}}}$$

When the result in Equation (30) is substituted into Equation (29), the impaction parameter *K* can now be calculated from readily available quantities.

The results obtained by Su and Yao [29] indicate that the transfer efficiency *ξ(K)* of impacting sprays onto the plate is a function of the impaction parameter. The predicted transfer efficiency is plotted against the impaction parameter *K* in Figure 1. It can be seen that almost all the points fall on the same curve. When *K* is greater than 0.3, the transfer efficiency is almost 1. In other words, if *K* exceeds the critical value of 0.3, according to [29], the droplets will be deposited; otherwise, they will be released into the air as overspray. The droplet diameter that corresponds to the critical value of *K* is called the critical diameter *d_crit_.*

Since a wall can be regarded as an infinite plate, the transfer efficiency *ξ(K_c_)* of the impacting spray can be defined as the ratio of the rate of droplets actually deposited onto the wall $\frac{dm_{Wd,l,c}\left(t_{k}\right)}{dt}$ to the overall rate $\frac{dm_{l,c}\left(t_{k}\right)}{dt}$ of the spray released in droplet size class *c* at the time $t_{k=l}$. For *k* ≠ l, $\frac{dm_{Wd,l,c}\left(t_{k}\right)}{dt}$ and $\frac{dm_{l,c}\left(t_{k}\right)}{dt}$ are assumed to be zero. Based on ξ(*K_c_*), the mass flow of the overspray $\frac{dm_{os,l,c}\left(t_{k}\right)}{dt}$ and of the impacting droplets $\frac{dm_{Wd,l}\left(t_{k}\right)}{dt}$ is then obtained as:(31)$$\frac{dm_{Wd,l,c}\left(t_{k}\right)}{dt}=\frac{dm_{l,c}\left(t_{k}\right)}{dt}\cdot \xi \left(K_{c}\right)$$
(32)$$\frac{dm_{os,l,c}\left(t_{k}\right)}{dt}=\frac{dm_{l,c}\left(t_{k}\right)}{dt}\cdot \left(1-\xi \left(K_{c}\right)\right)$$

The substance-specific mass flow rate $\frac{dm_{Wd,i,l,c}\left(t_{k=l}\right)}{dt}$ can be obtained by multiplying the right sides of Equations (31) and (32) with the mass fraction $w_{i}\left(t\right)$ (see Equation (7)) of the respective component *i.* Summation over all droplet size classes results in the overall impacting flow of component *i* at time *t_k_*.
(33)$$\frac{dm_{Wd,i,l}\left(t_{k}\right)}{dt}=\sum_{c=1}^{N_{c}}\frac{dm_{Wd,i,l,c}\left(t_{k}\right)}{dt}$$

### 2.9. Evaporation from Sprayed Walls

The mass balance equations for substance evaporation from a sprayed rigid surface can be formulated along the lines of evaporation from the floor. However, in contrast to the evaporation from liquid films deposited on the floor, spraying onto walls leads to an increase in the applied area that results in an increasing evaporation rate. As suggested in previous work [13], a logical approach to this problem is to break down the application area *W* into a sequence of *N_l_* small area elements, where each area element of size $\Delta W=W/\;N_{l}$ has its individual emission rate. The overall emission rate is then the sum of the individual emission rates.

For the differential mass balance of a single area element *l*, two mass flows must be considered for each component *i*, the impacting mass flow $\frac{dm_{Wd,i,l}\left(t_{k}\right)}{dt}$ and the evaporation rate $\frac{dm_{Wev,i,l}\left(t_{k}\right)}{dt}$. The differential mass change $\frac{dm_{W,i,l}\left(t_{k}\right)}{dt}$ of a component *i* in the liquid film on the wall at the time $t_{k}$ is then the difference of these flows:(34)$$\frac{dm_{W,i,l}\left(t_{k}\right)}{dt}=\frac{dm_{Wd,i,l}\left(t_{k}\right)}{dt}-\frac{dm_{Wev,i,l}\left(t_{k}\right)}{dt}$$
where analogously to Equation (25), the evaporation rate for an area element *l* can be expressed as
(35)$$\frac{dm_{Wev,i,l}\left(t_{k}\right)}{dt}=\Delta W\cdot \beta_{i}\cdot \left(\frac{M_{i}\cdot p_{i}^{*}\cdot x_{W,i,l}\left(t_{k}\right)\cdot \gamma_{i}\left(x_{W,i,l}\left(t_{k}\right)\right)}{R\cdot T}-C_{room,i}\left(t_{k}\right)\right)$$

If the evaporation rate is higher than the impaction flow, potentially leading to numerical results where $m_{W,i,l}\left(t_{k}\right)\leq 0$, the evaporation flow is assumed to be determined by the impaction flow, that is, $\frac{dm_{Wev,i,l}\left(t_{k}\right)}{dt}=\frac{dm_{Wd,i,l}\left(t_{k}\right)}{dt}$. In other words, the droplets are assumed to evaporate instantaneously when reaching the wall.

Using Euler’s method to estimate $m_{W,i,l}\left(t_{k}\right)$, $x_{W,i,l}\left(t_{k}\right),$ and $C_{room,i}\left(t_{k}\right)$, the total evaporation rate can then be calculated by summation over all area elements *l*:(36)$$\frac{dm_{Wev,i}\left(t_{k}\right)}{dt}=\sum_{l=1}^{N_{l}}\frac{dm_{Wev,i,l}\left(t_{k}\right)}{dt}$$

It should be noted that Equation (36) together with Equations (21) and (25), which are related to evaporation from aerosol droplets and from the floor, constitute the overall evaporation rate needed as indicated in Equation (3). Since evaporation from droplets is faster than evaporation from flat surfaces, the temporal resolution used in the Euler algorithm can be much lower when calculating the evaporation rate of flat surfaces. This fact was considered in the developed software by switching to longer discretisation times Δ*t_post_* if no aerosol generation had taken place (post-spraying phase) and exposure was dominated by evaporation from flat surfaces (wall, floor). In this way, the computing time could be significantly reduced.

## 3. Results

### 3.1. Example Calculation

Due to its flexibility, the proposed model algorithm potentially covers a range of exposure situations including the spraying of mixtures. One area where spraying is often used as an application technique is the disinfection of rooms and surfaces with biocidal products. In order to demonstrate the possibilities and limitations of the SprayEva algorithm, an example calculation is presented in the following with hydrogen peroxide as an active substance.

According to the Regulation (EU) No 528/2012 product assessment report [30], there are two product types amongst others, PT 2, 3, and PT 3 involves the disinfection of animal housing by spraying aqueous solutions of 7.4% (*w*/*w*) of hydrogen peroxide. This type of disinfection is typically performed by spraying aqueous hydrogen peroxide solutions with different devices generating larger droplets, leaving the product on the treated surfaces and not rinsing it off.

The worker typically uses a high-performance spraying device with a handheld spray lance. According to Koch et al. [12], for spraying onto a surface, flat fan or hollow cone nozzles with a flow rate of 0.6 L/min and a pressure of 3 bar are used, and flat cone nozzles tend to generate coarse droplets. Analyses performed so far with other models and the existing results of measurements at workplaces have demonstrated that the droplet spectrum of the spraying method used has an impact on the exposure concentration [12]. In the example calculations, a mass median diameter of $d_{m}=250\;\mu m$ and a hollow cone nozzle was assumed. The number of droplet size classes characterising the over spray was set to $N_{c}$ = 5. The geometric standard deviation of the droplet distribution was fixed to 1.8 and was not considered by Koch et al. [12] to be a parameter that is subject to much variation.

The example calculation aimed to characterize a situation where the room volume is located at the lower end of possible sizes of animal housings (see Table 1). The product concentration *w_pr,i_*, the surface coverage *SC*, and the ventilation rate *Q_vent_* were assumed to follow the data given in the ECHA assessment report. Since settled droplets probably do not cover the whole floor in real workplaces if only one wall is sprayed, the area of the floor relevant for evaporation was assumed smaller than the total floor area *F*. This was accounted for by multiplying *F* by an arbitrary factor of ¾, resulting in correspondingly reduced evaporation rates from the floor. Other process- and scenario-related parameters such as the room height *h_room_,* release rate *rr*, sprayed area *S*, application time *t_ap_* and exposure time *t_expo_*, opening angle of spray cone 2 *θ*, distance nozzle-wall *s*, and nozzle diameter *D*_0_ were estimated using the data given in [12]. Using Equation (29) together with Figure 4, the critical diameter could be estimated as *d_crit_* = 202 µm.

The substance-specific input parameters, including the activity coefficients γ, needed for the model calculation were adopted from previous work [13] and are listed in Table 1. The reference temperature *T_r_* for droplet evaporation was estimated using Hubbard’s rule [23] in combination with the empirical equation proposed by Hinds [14] for calculating the temperature at the droplet surface *T_dr_*. Regarding surface evaporation, calculations are based on the room temperature *T_room_*. The initial vapour concentrations *C_init_* of water and hydrogen peroxide were assumed to be zero.

In addition to these physically based parameters, the quantities determining the temporal and spatial resolution of the iteration must also be specified. The example calculations were carried out using temporal resolutions of Δ*t* = 0.2 s for the spraying phase and Δ*t_post_* = 2 s for the post-spraying phase, resulting in a total calculation time of about 8 min. The multiple integer *IM* of Δ*t*, determining the spatial resolution, was set to *IM* = 20, which proved to be a good compromise between accuracy and computation time.

The proposed extended Euler algorithm was programmed using the Mathematica^®^ environment. Typical calculation times are in the range of seconds to minutes as long as the spraying times are in the range of seconds to few minutes. However, spraying times > 5 min can be time consuming in the current code version, resulting in calculation times even in the range of hours. In contrast, longer post-spraying phases are less demanding in terms of computing speed because Δ*t_post_* can be set much longer than Δ*t*. This is because evaporation from flat surfaces is slower than the evaporation from droplets.

Using the input from Table 1 and Table 2 in Table 3, the time-dependent modelled aerosol (total and inhalable) and vapour concentrations as well as the sedimentation and emission rates were determined separately for H_2_O and H_2_O_2_. For clarification, the diagrams also include the individual time-dependent curves of the release pulses and the corresponding time-averaged curves.

Starting with the aerosol concentration, the graphs show that the time courses for the spraying and post-spraying phases can be quite different. In particular, it becomes clear that exposure to H_2_O_2_ aerosol reaches relevant concentrations only during the spraying phase. Table 3, panel 1 also reveals an increase in the aerosol concentration over the whole spraying phase, which can be explained by an increase in the backpressure (see panel 6), leading to slower evaporation rates (panel 3) and longer droplet lifetimes respectively. This is also reflected in panels 2 and 3 showing the settling and evaporation rates of H_2_O and H_2_O_2_ as components of aerosol droplets.

Although the aerosol droplets are more or less restricted to the spraying phase, evaporation from sprayed surfaces takes place over much longer time periods. The temporal pattern of evaporation is again determined by the volatility of the components and by the resulting back pressure. The influence of the activity coefficients is not so pronounced. Briefly, there is a time shift to be expected in the release of H_2_O_2_ because H_2_O_2_ is less volatile than water, which is confirmed in Table 3, panels 4 and 5. It should be highlighted that this delay is enhanced if there are large surfaces involved, leading to a significant backpressure that delays evaporation. This effect is also confirmed in panel 6, which shows lower vapour concentrations of H_2_O_2_ at the beginning of the exposure period.

The delayed occurrence of the concentration peaks of H_2_O_2_ suggests that the worker should leave the room at least after the application phase to minimize exposure as according to the Regulation (EU) No 528/2012 assessment report of a biocidal product (family) [30] hydrogen peroxide is assigned the hazard statement H332, ‘harmful if inhaled’.

### 3.2. Comparison with Measured Data

Although this paper predominantly outlines the theoretical concept of the new approach, answering the question “How closely does the model approximate to real workplace exposures?” is essential as well. Since confirming observations increases the likelihood that the model is not fundamentally flawed, some model estimates were compared with measured data as part of this research. It proved to be difficult to find quality data for scenarios against which to compare SprayEva exposure predictions. However, Koch et al. [12] carried out a comparison of SprayExpo and ConsExpo 4.1 estimates with measurement data from a limited number of simulated spray experiments. The experiments were performed in a storage room (length: 11.2 m, width: 4.5 m, height: 4.0 m) under controlled conditions using different spraying devices and nozzles (flat fan nozzle, hollow cone nozzle, cold fogger, thermal fogger) characterized by different MMDs and release rates. The geometric standard deviation of the droplet distribution was fixed to 1.8 and was not considered by Koch [12] to be a parameter that is subject to large variation.

The applications were performed according to the same time scheme:5 min measurements prior to spraying;7 min spraying (4 × 1 min spraying with 1 min interruption in between);30 min follow-up measurements.

The spray droplets were released directly into the room without the interference of limiting surfaces, such that they could be simulated exactly with the program part ‘room spray’ in SprayExpo, ConsExpo and SprayEva.

In all these experiments, an aqueous formulation consisting of 0.2% dysprosium acetate and 1% sodium chloride was applied. While water is considered the volatile component (see Table 4), the inorganic dysprosium acetate and sodium chloride were assumed nonvolatile due to their extremely low vapour pressure. The data recorded by Respicon [31] and by an unspecified hygrometer were used for the analyses of the time aerosol concentration of the nonvolatile portion and the relative humidity (RH), respectively. After the termination of the experiment, the filters of the Respicon were analysed for the amount of trapped dysprosium, and this value was then extrapolated to calculate the total mass concentration of the nonevaporating ingredients. The measured relative humidity was used to calculate the vapour concentration of water in the room air. For comparison with model estimates, the time-weighted average over the spraying and post-spraying phases was calculated.

The input parameters such as release rate, temperature, initial humidity, and MMD as well as the individual measurements used for comparison with estimates from SprayEva are summarised in Table 4, where the inhalable aerosol and vapour concentrations are expressed in terms of the 37 min time-weighted average. It should be noted that the values for room temperature and initial humidity were provided via personal communication with Koch.

Figure 5 shows the scatter plot of the Respicon measurements for the nonvolatile portion versus the model estimates of SprayExpo, ConsExpo 4.1, and SprayEva. The diagonal represents the 1:1 line, i.e., where the tool estimate is identical to the corresponding measurement value.

Looking at the deviations between the model values and the Respicon measurements, significant differences between the three models can be observed. For SprayExpo, underestimations are found only at low concentrations below 0.5 mg/m^3^. In contrast, SprayEva tends to overestimate exposure below 0.1 mg/m^3^. This partial lack of agreement is also confirmed numerically by the absolute and relative biases calculated according to Hornung [32], who defined bias as the mean difference between predicted estimates and measured exposure on a logarithmic scale. While the overall bias_SprayExpo_ = −0.13 and the relative bias_SprayExpo_ = −12.6% of SprayExpo were slightly negative, a positive bias_SprayEva_ = 0.18 and relative bias_SprayExpo_ = 19.6% was found for SprayEva. At the same time, Pearson correlation coefficients between the measurement results and tool estimates were r_SprayExpo_ = 0.998 and r_SrayEva_ = 0.992, demonstrating excellent correlation. Although the correlation coefficient of ConsExpo was excellent as well (r_ConsExpo_ = 0.994), for some experiments, the model estimates deviated considerably from the measurement results. It seems that overestimation increases with decreasing concentrations. This visually apparent trend toward overestimation was also confirmed numerically by the overall bias (bias_ConsExpo_ = 1.31) and the relative bias (rel.bias_ConsExpo_ = 269.6%).

While the experimental and measurement conditions were appropriate for the Respicon aerosol measurements, this was only partially the case for the water vapour concentration measurements. Firstly, the accuracy of the hygrometer used was comparatively low (+/−5%), and secondly, the atmospheric pressure was not measured; that is needed for the calculation of the mass concentrations. As a substitute, the mean atmospheric pressure of 101325 Pa was therefore used for the calculations. That is, weather-related changes in air pressure are not taken into account in the calculations. Figure 6 shows the scatter plot of the water vapour measurements versus the model estimates for SprayEva only; vapour exposure was out of the scope of the SprayExpo and ConsExpo spray models.

Looking at the deviations between the model values and the measurement values, SprayEva tends to overestimate exposure (bias_SprayEva_ = 0.14, rel.bias_SprayEva_ = 15.5%). In addition, the correlation between the tool predictions of exposure and the measurement data is r_SprayEva_ = 0.904, good but not excellent. This partial lack of agreement may be at least partly explained by the measurement limitations described above. Secondly, SprayEva does not take into account the condensation of water vapour on surfaces, which may occur in reality. Taking these limitations into account, the results are nevertheless regarded as supportive and acceptable (due to the trend of conservative estimates) for the plausibility and predictive power of SprayEva.

Finally, it has to be noted that the simulation results presented in Table 4 were calculated using temporal resolutions Δ*t* in the range between 0.01 and 0.05 s for the spraying phase and 2 s for the post-spraying phase. The integer multiple *IM* for Δ*t* was set to 20, that is, the spatial resolution and the calculation time were reduced by a factor of 20, which proved to be a good compromise between accuracy and computation time.

## 4. Discussion

Due to the complex relationships between substance properties, process parameters, work practices, and ventilation, currently there is a lack of mathematical models to provide useful predictions of worker exposure for spraying operations, especially for the spraying of (semi)-volatile substances. That is, exposure modelling requires strong simplifications taking into account only the most influential input parameters. Since this inevitably leads to model uncertainties, there is always a trade-off between the accuracy of a model, the number and availability of the required input parameters, and the scope of application of the model. Because the availability of the input parameters is limited in the risk assessment procedure under the Biocide and REACH regulations, the main goal when developing SprayEva was to keep the model as simple as possible without losing too much predictive power and scope. For instance, we decided not to go beyond the well-mixed room model because published studies [18,33,34,35,36] and our own evaluation exercise revealed reasonable agreement with the measured data as long as room sizes were small to medium. However, exposure estimates assuming well-mixed room conditions may be questionable for larger rooms. In this case, more sophisticated models such as two-box models should be used to avoid the underestimation of exposure. It should be noted that the iterative approach of SprayEva can be readily extended to more compartments, taking into account technical control measures [37] such as local exhaust ventilation or containment. Finally, this study was restricted to binary mixtures because they lead to much simpler equations, highlighting the dominant parameters for droplet evaporation. At the same time, they allow for the much quicker calculation of the droplet evaporation since there are fewer equations to be solved. However, it should be noted that according to Wilms [22], mixtures with more than two components can addressed with SRMM as well. This can be a topic for future work.

This study primarily outlines the theoretical concept of the model without claiming to provide a complete empirical, measurement-based evaluation of the room- and wall-spraying scenarios covered by SprayEva. Unfortunately, suitable measurement data on vapour concentrations during wall-spraying scenarios are not available. Therefore, the data used for comparison with model estimates are restricted to the room spraying of an aqueous solution of nonvolatile components. Such exposure situations are within the scope of the proposed approach but are not typical for SprayEva, which mainly targets (semi)-volatile mixtures. While the comparisons of the aerosol concentrations of the nonvolatile components with exposure estimates showed good agreement, experimental evidence for accurate model estimates of SprayEva for (semi)-volatile components is still subject to uncertainty. Although some measurement data for water vapour are available, the informative value of the data is limited due to the nonoptimal measurement conditions. In particular, the lack of measuring accuracy of the hygrometer used should be mentioned here. Nevertheless, the correlations of the measured data with the model estimates are good and are only moderately positively biased, which might be explained by water vapour condensation on surfaces, which is not addressed by SprayEva. Thus, the model tends to overestimate vapour exposure, which is desirable from a safety perspective.

The example calculations of the spraying of H_2_O and H_2_O_2_ mixtures reveal that exposure to aerosol droplets is more or less restricted to the spraying phase. The evaporation from sprayed surfaces takes place over much longer time periods, as illustrated by panels 5 and 6 in Table 3, where the temporal pattern of evaporation is determined by the volatility of the components. In brief, there is a delay to be expected because H_2_O_2_ is less volatile than water, which causes water to evaporate first. The example calculations also show that evaporation from large surfaces creates a backpressure that in turn affects the time course of air concentrations by significantly delaying the release of substances. This suggests that the worker should leave the room after the spraying phase to minimize exposure. As already addressed in previous work [13], the nonideal behaviour of components should be considered in any case. Although nonideal behaviour is not very pronounced in the case of H_2_O_2_/H_2_O mixtures, it can play an important role with mixtures whose components differ greatly in polarity. Therefore, when using SprayEva, exposure assessors should be aware of deviations from ideal behaviour because the delayed or premature release of hazardous substances can determine the risks workers are facing.

Although the proposed mass balance equations in combination with the iterative approach provides a very flexible framework for modelling of exposure scenarios, there are still limitations the assessor has to consider when using SprayEva.

The strong simplification regarding the dispersion of the spray cloud by instantaneous diffusion differs heavily from the actual physical dispersion behaviour, in particular in high or very large rooms. In these settings, a concentrated particle cloud is initially created close to the source, spreading only very gradually through the whole room. Therefore, special conditions such as overhead spraying are not further differentiated in the well mixed room model. In addition, the impaction module is at present restricted to wall spraying and does not include spraying onto floor surfaces, where sedimentation and impaction contribute to the overall mass on the floor, making the mass balance difficult.

Furthermore, the influence of chemical reactions on the concentration course is not yet taken into account, which can play a significant role in practice (e.g., reaction of H_2_O_2_ with biofilms). Although these effects can in principle be considered in the mass balance to some extent, the corresponding kinetics are regarded as complex and might be a topic for future research. This also applies to more sophisticated dispersion models, such as two-box models or models that can take into account additional technical control measures, such as local exhaust ventilation or containments. For instance, Ganser and Hewett [37,38] demonstrated that one- and two-box models can be extended to situations where various forms of a local exhaust ventilation are involved.

Finally, it has to be noted that calculations with the current code in the Mathematica^®^ [39] environment can be time consuming since the computing time increases quadratically with the number of iteration steps. For the post-spraying phase, computing speed is less demanding because the evaporation process from flat surfaces is slower than for droplets, allowing for longer discretisation intervals. Computer capacity and code optimization is therefore a crucial factor for reducing the computing time. First attempts in this direction using an R-environment showed that calculation time could be reduced by more than an order of magnitude.

## 5. Conclusions

An important finding from the example calculations and from the evaluation exercise is that the proposed iterative algorithm is very flexible, covers a wide range of scenarios including binary (semi)-volatile mixtures, and allows variable application patterns in room- and wall-spraying scenarios.

The example results suggest that situations where high evaporation rates from large surfaces and hence backpressure occur can influence the time course of airborne concentrations by significantly delaying substance release.

This also applies for mixtures where liquid-phase nonidealities play a role. Although nonideal behaviour is often less pronounced in practice, mixtures whose components differ greatly in polarity should be paid special attention, as deviations from ideal can lead to the delayed or early release of components. Both deviations from ideal and backpressure effects can significantly change the time course of exposure, leading to exposure after the original application is finished and thus possibly requiring the adjustment of risk management measures.

Finally, it has to be noted that the relationships between substance properties, process parameters, work practices, and ventilation are complex and require strong simplifications in model building. Therefore, exposure estimates are inevitably uncertain to some extent. Nevertheless, the small-scale evaluation exercise showed good to excellent correlations between measured data and model estimates and only a small to moderate tendency towards overestimation. Thus, the results are regarded as supportive for the predictive power of SprayEva and acceptable from a safety perspective. Promising applications of the new model are, for example, spray applications of disinfection and cleaning products as they often contain volatile and (semi)-volatile active substances.

## Figures and Tables

**Figure 1 ijerph-19-13182-f001:**
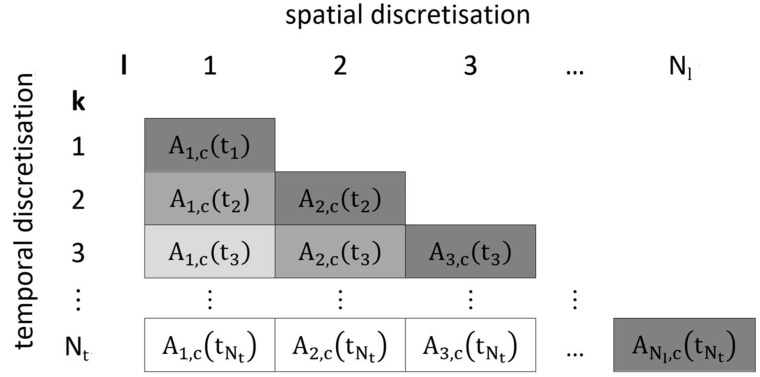
Discretisation approach for *IM* = 1.

**Figure 2 ijerph-19-13182-f002:**
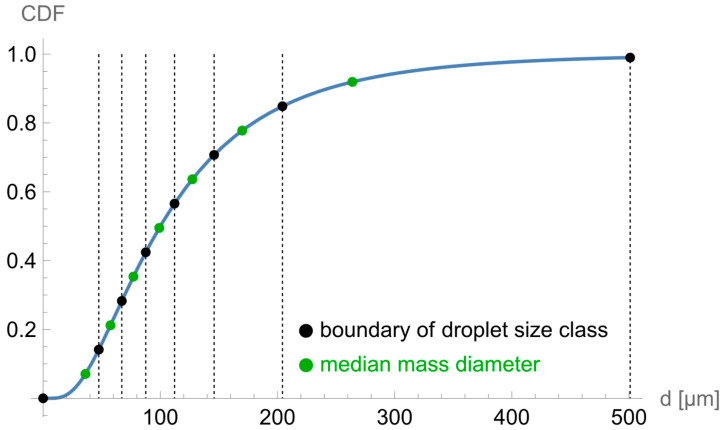
CDF of a cumulative mass distribution.

**Figure 3 ijerph-19-13182-f003:**
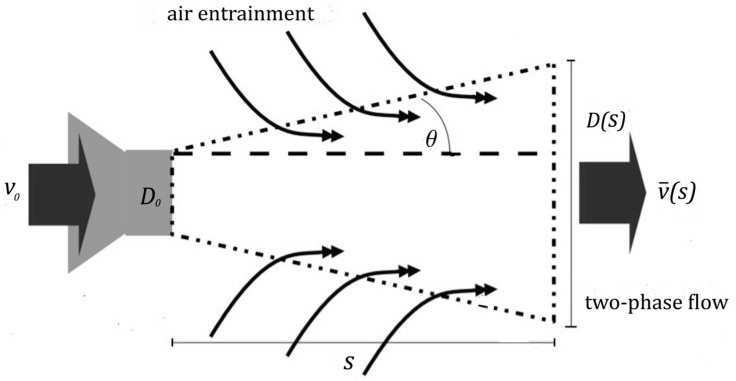
Diagram of the mass entrainment process (adopted from [28]).

**Figure 4 ijerph-19-13182-f004:**
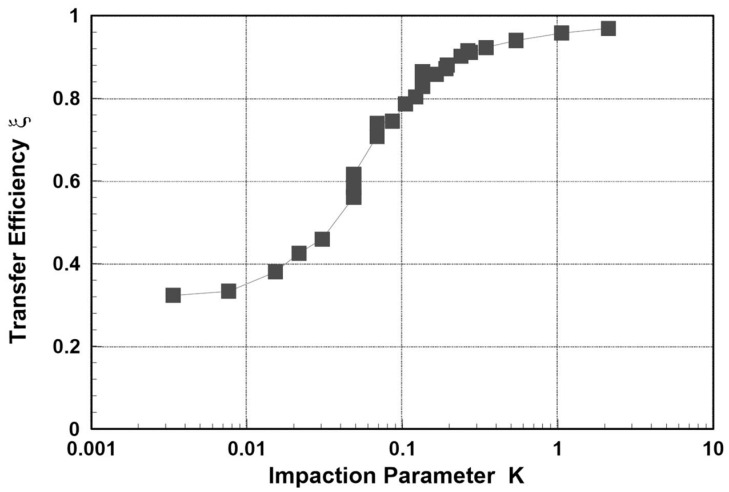
Spray transfer coefficient ξ against the impaction parameter *K* (adopted from [29]).

**Figure 5 ijerph-19-13182-f005:**
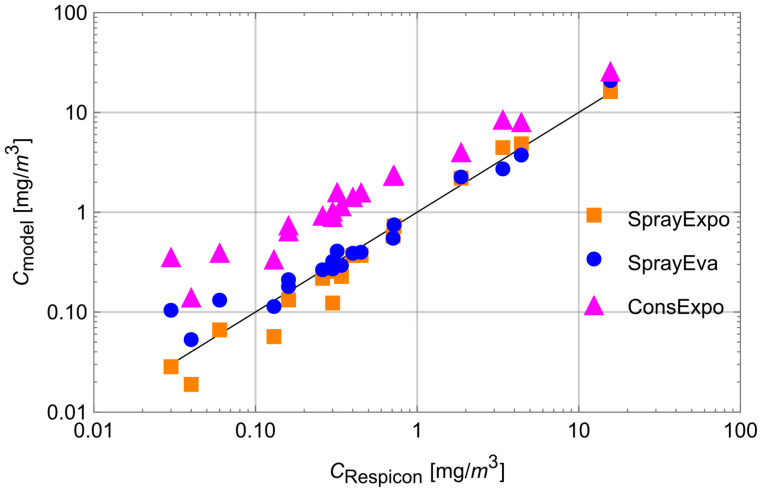
Measured data vs. SprayExpo, ConsExpo, and SprayEva estimates of aerosol exposure.

**Figure 6 ijerph-19-13182-f006:**
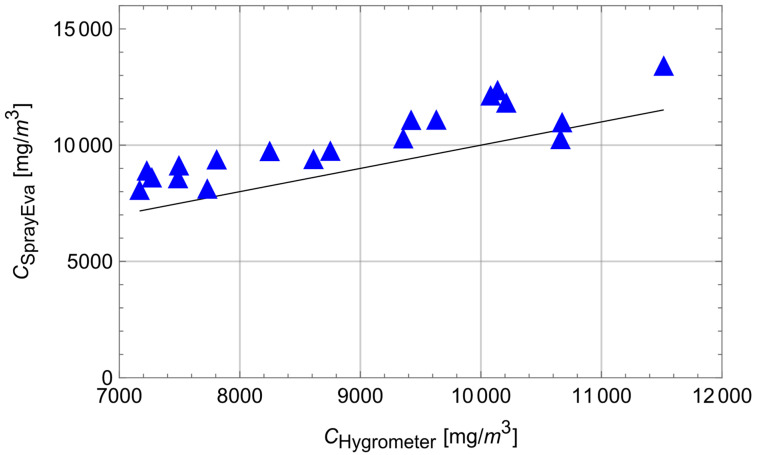
Measured data vs. SprayEva estimates of H_2_O-vapour exposure.

**Table 1 ijerph-19-13182-t001:** Scenario-related parameters.

*T_room_* = 293 K	*T_dr_* = 283.8 K
*V_room_* = 150 m^3^	*h_room_* = 2.5 m
*Q_vent_* = 0.125 m^3^/s	*s* = 0.4 m
*ν_air_* = 0.5 m/s	$C_{init}$ = 0 mg/m^3^
*t_app_* = 300 s	*t_expo_* = 18,000 s
*W* = 20 m^2^	*SC* = 0.06 kg/m^2^
*rr* = 8 g/s	*θ* = 25 deg
*d_m_* = 250 µm (hollow cone)	GSD = 1.8
*D*_0_ = 10^−3^ m	v_0_ = 12.7 m/s

**Table 2 ijerph-19-13182-t002:** Substance-specific input parameters.

*p*_H2O_ =* 2308 Pa at 293 K	*p*_H2O2_* = 179 Pa at 293 K
*p*_H2O_* =1276 Pa at 283.8 K	*p*_H2O2_* = 88 Pa at 283.8 K
*w_pr,H2O_ =* 0.926	*w_pr,H2O2_ =* 0.074
*D_H2O_* = 2.4 × 10^−5^ m^2^/s	*M_H2O_*=18 × 10^−3^ kg/mol
*D_H2O2_* = 1.8 × 10^−5^ m^2^/s	*M_H2O2_*=34 × 10^−3^ kg/mol
*β_H2O_* = 2.4 × 10^−3^ m/s	*β_H2O2_* = 2.2 × 10^−3^ m/s
*η* = 1.82 × 10^−5^ kg/(m∙s)	ν = 1.53 × 10^−5^ m^2^/s
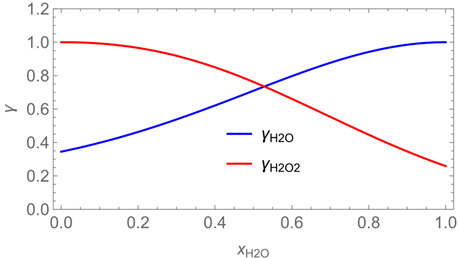	

**Table 3 ijerph-19-13182-t003:** Time-dependent modelling results for different parameters of interest.

№	H_2_O	H_2_O_2_
1	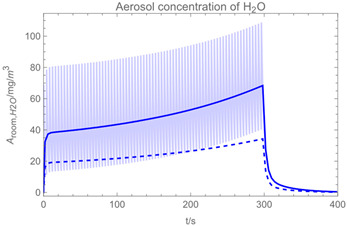	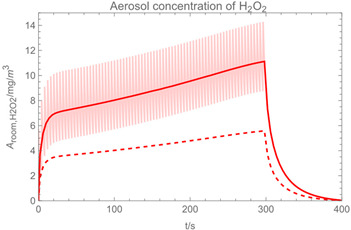
2	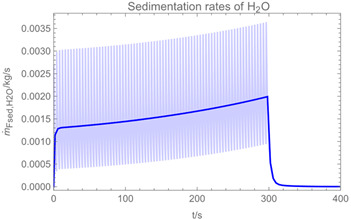	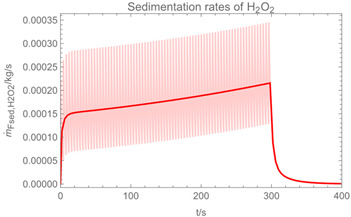
3	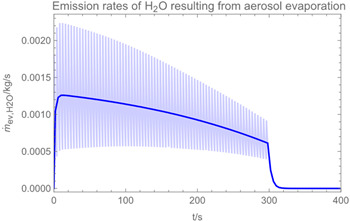	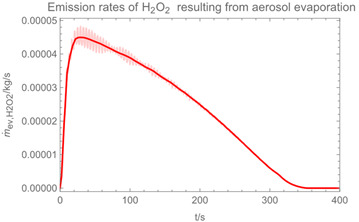
4	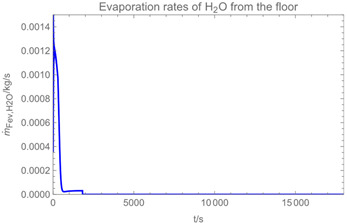	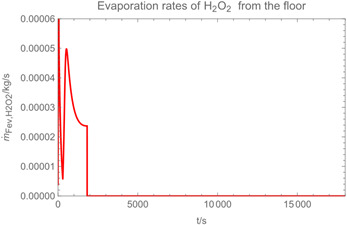
5	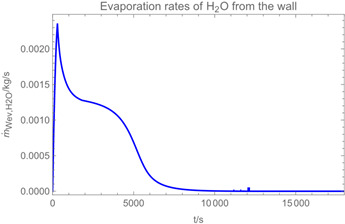	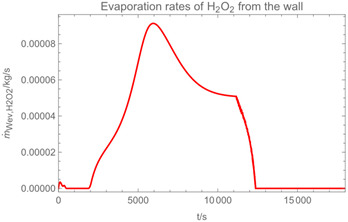
6	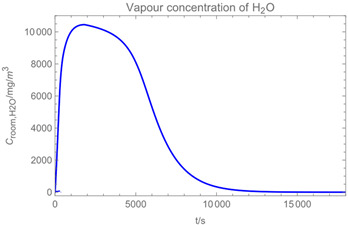	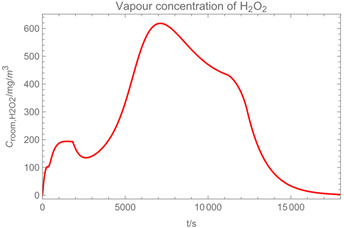
Legend	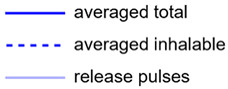	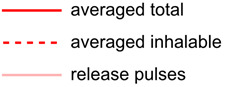

**Table 4 ijerph-19-13182-t004:** Input parameters and measured values vs. model estimates.

Type of Nozzle	Release Rate	MMD	Temp.	Init. RH	Respicon	SprayEva	Hygrometer	SprayEva
	[g/s]	[µm]	°C	%	Inhalable Fraction [mg/m^3^]		H_2_O Vapour [mg/m^3^]	
Flat fan	19.6	400	16.1	38	0.32	0.43	7227	9031
Flat fan	15.4	470	16.3	38	0.3	0.29	7729	8252
Flat fan	10.0	600	16.6	42	0.03	0.11	7168	8200
Flat fan	5.0	250	16.7	44	0.3	0.34	7494	9250
Flat fan	4.2	400	17.0	44	0.13	0.12	7487	8700
Flat fan	2.7	500	17.5	45	0.04	0.056	7269	8735
Flat fan	10.4	340	18.3	39	0.34	0,31	8247	9870
Flat fan	8.3	380	17.7	43	0.16	0.22	7807	9510
Flat fan	5.4	420	19.3	43	0.06	0.14	8750	9880
Hollow cone	14.2	280	18.0	43	0.71	0.58	9630	11,220
Hollow cone	11.3	300	17.9	40	0.45	0.42	9421	11,210
Hollow cone	6.0	290	18.1	40	0.26	0.28	8611	9525
Hollow cone	9.8	230	18.2	42	0.72	0.79	10,673	11,110
Hollow cone	8.1	270	17.7	45	0.4	0.41	10,660	10,385
Hollow cone	5.8	340	19.2	45	0.16	0.19	9356	10,415
Cold fogger	5.5	130	20.1	45	1.87	2.32	10,210	11,954
Cold fogger	13.3	140	19.0	40	3.39	4.69	11,516	13,540
Cold fogger	7.9	110	18.6	42	4.41	5.12	10,138	12,488
Therm. fogger	7.3	50	21.4	33	15.7	17.1	10,080	12,261

## Data Availability

Data is contained within the article. The data presented in this study are available in [12].

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
