# Peer review of "A New Model Algorithm for Estimating the Inhalation Exposure Resulting from the Spraying of (Semi)-Volatile Binary Liquid Mixtures (SprayEva)"

_ijerph, 2022, doi:10.3390/ijerph192013182_

Round 1

Reviewer 1 Report

In this work a new model algorithm for estimating inhalation exposure resulting from spraying of (semi)-volatile binary liquid mixtures is described. The proposed iterative algorithm is very flexible, covers a wide range of scenarios, and allows variable application patterns in room- and wall-spraying scenarios. The work is of interest because the results are regarded as supportive for the predictive power of SprayEva and acceptable from a safety perspective. The article looks like a short communication and may be published after minor revision.

Notes:

1. From what point of view authors did chose binary mixtures for investigations? Is this new model algorithm for estimating inhalation exposure suitable for multicomponent mixtures?

2. Conclusion should be shortened and focused on the main findings of this work.

3. Promising application of new obtained results should be added in Conclusion.

Author Response

Thank you for your review. We appreciate your comments and would like to implement them in the manuscript as follows:

Response to point 1: We propose to include the following text in the discussion section in order to highlight why a binary mixture was chosen for investigation instead of accounting for multicomponent mixtures: “This study was restricted to binary mixtures because they lead to much simpler equations highlighting the dominant parameters for droplet evaporation. At the same time they allow a much quicker calculation of the droplet evaporation, since there are fewer equations to be solved. But it should be noted that according to Wilms [22] mixtures with more than 2 components can addressed with SRMM approach as well.”

Response to point 2: Agreed, we propose to delete the following sentences: “Based on an extended Euler algorithm, with temporal and spatial discretization, numerical solutions of the underlying differential equations can be obtained for a variety of spraying applications of binary semi-volatile mixtures. For wall-spraying scenarios an impaction module has been adopted that quantifies the amount of overspray and the amount of material that lands on the wall. It includes situations where the moving workers apply the product continuously, with increasing surface area over time from which evaporation takes place. Time dependent concentration gradients within the sprayed liquid films, that may result from different volatilities of the components, are therefore addressed by the proposed model. With this in mind, risk assessors should also consider the impact of back pressure.”

Response to point 3: Agreed, we propose to include the following sentences in the Conclusions: “Promising applications for the new model are for example spray applications of disinfection and cleaning products as they often contain volatile and semi-volatile active substances.”

Reviewer 2 Report

In the paper the previous model developed by the authors to quantify the exposure in the air is adapted or improved. The model is based on ODEs and takes into account different aspects of the problem.

The paper is well-written and the model is described in detail. Finally, in Results an application of the model is presented. In my opinion, the paper should be accepted and I have only two comments.

1. The abstract should be shortened. The current version is too long.

2. On page 17 is written This was accounted for by multiplying F with an arbitrary factor of 3/4. How affects the choice of this factor in the model?

Author Response

Thank you for your review. We appreciate your comments and would like to implement them in the manuscript as follows:

Response to point 1: Agreed, we propose to delete the following sentences: “Given the large number and variety of chemical substances present in spray products and the variability in use patterns, it is not possible to measure all potential exposures, but exposure modelling is imperative as well. Although several models are available in legal contexts, at present there are no models that can simulate exposures for spraying of mixtures of (semi)-volatile components…. Since in the case of wall-spraying only the non-impacting fraction of the droplet spectrum (overspray) is relevant for aerosol exposure, an impaction module has been developed that aims to quantify the amount of overspray and the amount of material that lands on the wall….. Looking at the deviations between the model values and the measurement values of the Respicon, significant differences between the four models can be observed.” A shortened sentence has been introduced reflecting the role of wall-spraying scenarios: “For wall-spraying scenarios an impaction module has been developed that quantifies the amount of overspray and the amount of material that lands on the wall.” Thus, the abstract is shortened by 670 characters.

Response to point 2: The introduction of the factor F is intended to represent the fact that spraying of only one wall in an animal house does not cover the entire floor with settled droplets. Considering a factor of 3/4 leads to a correspondingly reduced evaporation rate of settled material from the floor in the model. We suggest the following change in the text: “This was accounted for by multiplying F with an arbitrary factor of ¾, resulting in correspondingly reduced evaporation rates from the floor.”
